# Efficacy of a New Hemostatic Dental Sponge in Controlling Bleeding, Pain, and Dry Socket Following Mandibular Posterior Teeth Extraction—A Split-Mouth Randomized Double-Blind Clinical Trial

**DOI:** 10.3390/jcm12144578

**Published:** 2023-07-10

**Authors:** Armin Mahmoudi, Mohammad Ali Ghavimi, Solmaz Maleki Dizaj, Simin Sharifi, Seyyede Shabnam Sajjadi, Amir Reza Jamei Khosroshahi

**Affiliations:** 1Department of Oral and Maxillofacial Surgery, Faculty of Dentistry, Tabriz University of Medical Sciences, Tabriz 51548-53431, Iran; arminm86@yahoo.com (A.M.);; 2Dental and Periodontal Research Center, Tabriz University of Medical Sciences, Tabriz 51548-53431, Iran; 3Department of Dental Biomaterials, Faculty of Dentistry, Tabriz University of Medical Sciences, Tabriz 51548-53431, Iran; 4Department of Pediatric Dentistry, Faculty of Dentistry, Tabriz University of Medical Sciences, Tabriz 51548-53431, Iran

**Keywords:** gelatin, sponge, hemostatic, pain, dry socket, randomized, double-blind, split-mouth

## Abstract

Aims: This study aimed to clinically evaluate of a novel gelatin-based biodegradable sponge after mandibular posterior teeth extraction to assess its abilities in controlling bleeding, pain, and dry socket compared a commercial sponge. Trial design: In this study, 26 patients who needed the extraction of two mandibular molar teeth were selected and, in each patient, after tooth extraction, the prepared gelatin sponge was used in the test group and the commercial sponge was used in the control group in the form of a randomized, double-blind, split-mouth clinical trial. The sterile gauzes were used on top of each sponge to absorb the extra blood (unabsorbed blood of sponges) to assess the blood absorption amount. Also, the amount of bleeding was recorded for 1 and 4 h after extraction for two groups. The amount of pain was measured for 12, 24, and 48 h after tooth extraction by Visual Analogue Scale (VAS). All patients also returned for examination four days after extraction to assess the occurrence of dry socket. Results: The results showed that the average weight of absorbed blood by sterile gauze in the control group (6.32 ± 1.06 g) was higher than in test group (3.97 ± 1.1 g), e.g., the bleeding control was better for the test group (*p* < 0.05). Bleeding was observed to be significantly reduced in the test group within 1 h (*p* = 0.003), within 1–4 h (*p* = 0.002), and after 4 h (*p* = 0.042) post-operatively in comparison to the control group. The average pain decreased significantly over time in both groups and the reduction of the pain was significantly higher for the test group (*p* < 0.05). Just one dry socket case occurred in the control group. Conclusion: The prepared sponge is recommended for use in dental surgeries because of its abilities in bleeding, pain, and dry socket control.

## 1. Introduction

Tooth extraction is one of the most common dental treatments associated with several known complications. These complications may occur during or after surgery [1,2]. Bleeding, pain, and infection are early complications after tooth extraction [3,4,5], and the incidence of these complications is based on risk factors such as the patient’s age, gender, underlying diseases, the use of certain medications, the location of the tooth, smoking, and poor hygiene. Mouth, number of operated teeth, anesthesia technique, use of antibiotics before surgery, use of drugs inside the cavity, and the experience of the surgeon are variable [6,7]. Bleeding and pain are common complications after surgery [8,9]. Continued bleeding without forming a clot, or prolonged bleeding for 8 to 12 h, which is known as post-extraction bleeding (PEB), is abnormal [10].

After tooth extraction, an open wound is left in the soft tissue of the bone and impaired wound healing is one of the most concerning problems after tooth extraction [11]. Often, the use of corticosteroid drugs and steroid anti-inflammatories is prescribed locally and systemically to control pain and bleeding [12]. Ice compresses, antibiotics, and even low-power lasers and hypothermia are also used [13,14,15]. None of these methods are without complications, so it is necessary to find a suitable method with few or no complications.

A dry socket or dry cavity is a complication in dentistry that sometimes occurs after tooth extraction and is caused by the non-formation of a blood clot [16,17,18]. In our previous study, we studied the use of gelatin porous sponge in reducing or improving dry sockets. According to the results obtained from that study, the possibility of reducing the prevalence of dry socket by using sponge inside the extracted tooth cavity can be explained due to the retention of the clot by this material inside the extracted tooth cavity. According to the results, the prevalence of dry sockets in the experimental group showed a decrease compared to the control group [19].

Gelfoam is a porous gelatin sponge, insoluble in water, flexible and absorbable, which is prepared from animal skin gelatin granules [20]. Gelfoam has a hemostatic nature and is also used to control bleeding after tooth extraction surgery [21]. It is used dry or saturated with sterile saline solution. These sponges are used by keeping the blood in the form of clots, protecting the wound, increasing the speed of recovery, and reducing complications after extraction [22].

Gelatin is a widely used macromolecule that has been accepted as a safe substance by the US Food and Drug Administration (FDA) and complies with the US, European, and Japanese pharmacopeias. Of course, this useful and safe material has drawbacks, such as low mechanical strength and weak resistance to hydrolysis. These defects can be solved by cross-linking gelatin using physical, chemical, or enzymatic approaches [23]. In addition, the use of gelatin as a hemostatic agent has been reported to be useful in accelerating the conversion of fibrinogen to fibrin during the coagulation process and also promoting fibrin polymerization. Also, gelatin has a very special homeostatic effect due to its inherent characteristics, which include non-immunogenicity, biocompatibility, biodegradability, and accessibility. Gelatin has been used in different industries and in different formulations for diverse applications, including continuous drug release and tissue engineering. Another characteristic of this material is that it can activate platelets and acts as an absorbent material. However, because gelatin is easily dissolved in water, the investigators use some methods such as mixing it with other natural and synthetic polymers to improve its mechanical and chemical stability [24].

An ideal hemostatic material should have conditions such as lightness, stability, and the ability to be easily inserted into the bleeding site. Furthermore, a hemostatic material should be flexible enough to conform to the wound to ensure that they reach and apply pressure to damaged areas that are otherwise inaccessible. An ideal hemostatic material should also not cause tissue destruction, not disintegrate into particles entering the bloodstream, be stable enough to withstand the high pressures affected by bleeding in the vessels and be easily removed after the bleeding has stopped [24].

In the present study, we compared the abilities of a new gelatin sponge compared with a commercial Gelfoam in controlling bleeding, pain, and the incidence of dry socket after split-mouth extraction of mandibular molars in patients in the form of a double-blind clinical trial. This new material synthesized was a new hemostatic dental sponge in our previous study [19]. It was non-toxic against human fetal foreskin fibroblasts (HFFF2) cells and red blood cells (RBCs). It showed the efficiency of more than two conventional commercial dental sponges as well [19].

## 2. Materials and Methods

### 2.1. Materials

Carbopol containing 2% lidocaine and 1: epinephrine was prepared from Septodont Healthcare, Maharashtra, India. Commercial Gelfoam was purchased from Spongostan^®^; Ferrosan, Soe-borg, Copenhagen, Denmark. Sterile gauze was prepared from Dermarose, Tehran, Iran.

### 2.2. Preparation of Dental Gelatin Sponge

The sponge used in this study was prepared and physiochemically characterized in our previous study [19]. The aeration method of gelatin aqueous solution was used to prepare the hemostat sponge. For this purpose, a gelatin aqueous solution (10%) was prepared. Then, 2% glycerol and 1% crosslinker agent (Glutaraldehyde) were added. Finally, the prepared thickened material was placed at −80 °C for 24 h, and then a freeze dryer was used for drying the material. The prepared materials were cut and sterilized via gamma irradiation (25 kGy). The properties of the produced sponge are shown in Figure 1. The SEM image displayed a porous structure with nanometric and micrometric pores. Nanometric pores can assist in increased plasma sorption, and micrometric pores can be applied for tricking red blood cells (RBCs) [25]. Nanometeric pores on the surface of the sponge can also lead to surface roughness [26]. The existence of carbon, nitrogen, and oxygen in the EDX spectra was related to gelatin [27].

### 2.3. Dates for Trial

Expected recruitment start date: 19 May 2022

Expected recruitment end date: 20 July 2022

Expected recruitment start date means the start date of the study steps (preparing materials, preparing patients, the main clinical process).

The expected recruitment end date of the study means the end of the main clinical process, concluding, statistical analyzing, writing the final report and so on.

The main clinical process was 4 days including the tooth extraction process, sponge insertion, blood absorption evaluation, the amount of bleeding and pain evaluation (one day evaluation), and controlling any signs and symptoms of infection or dry socket four days after tooth extraction process.

### 2.4. Calculation of Sample Size

The main criterion for determining the sample size was the hemostatic effect (blood clotting). The amount of pain and the incidence of dry socket are among the sub-goals of the plan, which are carried out in order to follow up on the patient’s condition. In order to determine the sample size, the hemostatic effect was obtained based on the results of Piri et al.’s study [28] in the control and test groups, respectively, 50.7% and 97.2%. Considering the first type error equal to 0.05 and the power of 80%, 26 subjects were selected as the sample size.

### 2.5. Randomizing and Blinding

Patients were randomly allocated into the test group and control group in accordance with the lottery method with a 1:1 distribution ratio. The blinding was double-blind with both the assessors and patients blinded. A.J. made the random distribution order. A.M. registered participants and assigned participants to involvements.

### 2.6. Population

This double-blind split-mouth clinical trial study was directed on 26 patients referred to the Department of Oral and Maxillofacial Surgery, Faculty of Dentistry, Tabriz University of Medical Sciences, Tabriz 51548-53431, Iran for the extraction of two mandibular molar teeth (first and second), taking into account compliance with the inclusion and exclusion criteria of the study. Table 1 shows the demographic details of the study subjects.

### 2.7. Inclusion Criteria

The need for extraction of two mandibular molar teeth (first and second),

Age between 20–45 years,

Not use of corticosteroids,

Not use of antibiotics in the last month,

Not coagulation disorders,

Not acute or uncontrolled infection in the dental surgery site,

Not malignancy in the dental surgery site,

Not exposure to the site Radiation surgery,

Not taking oral contraceptives in the last month,

Not smoking,

Not having hypertension, diabetes, thyroid disorders, and blood diseases.

### 2.8. Exclusion Criteria

People who use anti-coagulant drugs,

Not signing the consent form,

Suffering from a systemic disease,

Patients whose teeth cannot be extracted in the usual way and there is a need for surgery and section.

### 2.9. The Tooth Extraction Process

The removal of mandibular third molars is often accompanied by significant postsurgical sequelae and different procedures have been defined to decrease such adverse happenings. Standard extraction processes were arranged in the study and control groups. The test side was selected using the lottery method and the contralateral side was considered as a control group. The extraction of all teeth was done under the same conditions and by the same dentist. All surgical tools and techniques were the same for all patients. Necessary training and recommendations after tooth extraction in both stages of tooth extraction were provided to the patients by a dental specialist specializing in maxillofacial surgery. The usual recommendations included avoiding smoking, avoiding spitting, and using any ready made mouthwash to prevent interference with the treatment. Carbopol containing 2% lidocaine and 1% epinephrine was used for anesthesia for all patients. If the teeth were not removed in the usual way and there was a need for surgery and section, the patient was excluded from the study to reduce the role of confounding factors (for example trauma during extraction and the method of extraction).

### 2.10. Sponge Insertion

In each patient, after tooth extraction, a newly developed gelatin sponge was placed in the socket of the extracted tooth for dressing, then a sterile gauze with a specific weight was placed on it (test group). In the other tooth (split-mouth), a commercially available Gelfoam (Spongostan^®^; Ferrosan, Soe-borg, Copenhagen, Denmark) was placed in the dental socket, then a sterile gauze with a specific weight was placed on it (control group). Figure 2 shows the sponge insertion process.

### 2.11. Blood Absorption Evaluation

In this evaluation, the amount of blood clotting for a new gelatin sponge was compared with a commercially available Gelfoam (Spongostan^®^; Ferrosan, Soe-borg, Copenhagen, Denmark) in two extracted teeth. In the studied groups, sterile gauze was changed every 15 min until the bleeding stopped completely. The difference in the weight of the sterile gauze at the beginning and after the absorption of blood showed the amount of blood absorption of each sterile gauze. The number of sterile gauzes used, and the total weight of sterile gauze absorbed by the blood in each group (test and control) were compared.

### 2.12. The Amount of Bleeding

Also, the amount of bleeding after the removal of sterile gauze (within one hour after tooth extraction) and after the removal of anesthesia (within 1–4 h after tooth extraction) were recorded for two groups in each patient. The amount of uncontrollable bleeding after 4 h was also considered active bleeding and was checked and recorded. The bleeding index was calculated clinically and had grades 0 to 4: grade 0 (very low), grade 1 (low), grade 2 (normal), grade 3 (high), and grade 4 (very high) [29].

### 2.13. Pain Evaluation

The amount of pain was measured in the first 12 h after tooth extraction, as well as at 24, 48, and 72 h after tooth extraction. The pain was measured using a Visual Analogue Scale (VAS). The VAS consists of a 10 cm line, with two end points representing 0 (‘no pain’) and 10 (‘pain as bad as it could possibly be’). We asked the patients to degree their existing scale of pain by inserting a mark on the line [30].

### 2.14. Occurrence of Dry Socket

All patients also returned for examination 4 days later and were examined for the occurrence of dry socket. The patients were checked on signs and symptoms of infection or dry socket including severe pain within 4 days after a tooth extraction, observable bone in the socket, fractional or entire loss of the blood clot at the tooth extraction site, pain that releases from the socket to ear, eye, temple, or neck on extraction side of face, bad breath, nasty taste in the mouth [31].

### 2.15. Statistical Analysis of Data

Demographic information of the study participants was shown using descriptive statistics. In order to compare the amount of bleeding between the studied groups, Fisher’s exact test was used, the Mann–Whitney test was used to compare the pain level in the two groups, and descriptive statistics were used to express the percentage of dry socket in each group. SPSS version 19 software was used for data analysis. The *p* < 0.05 was considered a significant level.

## 3. Results

Twenty-nine subjects were selected for the study based on inclusion and exclusion criteria but three were omitted as they were lost to follow-up, so the net sample size was 26 subjects as shown in the CONSORT flow diagram (Figure 3). As the study was a split-mouth randomized clinical trial, one side (left) of the CONSORT flow diagram is related to the test group and the other side (right) is related to the control group.

### 3.1. Blood Absorption

Table 2 demonstrates the outcomes of blood absorption, including the number of used sterile gauze. The results of Table 2 show that there was no significant difference in the number of gauzes in the two groups and in both groups, consumption of three gauzes was the most frequent.

In addition, the average weights of absorbed blood by sterile gauzes in the control and test groups were 6.32 ± 1.06 g and 3.97 ± 1.1 g, respectively (*p* < 0.05).

### 3.2. The Amount of Bleeding

Bleeding was observed to be significantly reduced in the test group within 1 h (*p* = 0.003), within 1–4 h (*p* = 0.002), and after 4 h (*p* = 0.042) post-operatively in comparison to the control group. Table 3 shows the results of bleeding for both groups. The total weight of absorbed blood by a sterile gauze was higher in the control group, displaying the greater ability for blood absorption of the test sponge.

### 3.3. Pain Evaluation

The results of Table 4 show that in both groups, according to Friedman’s test, the average pain decreased significantly over time (*p* = 0.041). Also, according to the Mann–Whitney test, there was a significant difference in average pain between the two groups at any of the investigated times (*p* = 0.032).

### 3.4. Occurrence of Dry Socket

In the test group there were no cases of dry socket whereas there was one case of a dry socket (3.7%) observed in the control group (commercial sponge) after four days.

## 4. Discussion

In recent years, absorbable sponges have been described to show good results as progress in tooth extraction and oral surgeries. It is reported that these sponges avoid postoperative problems [22,32]. Walter et al. used commercial Gelfoam (Pfizer Co., Kalamazoo, MI, USA) in their study to report data on a series of 250 cases needing tooth extraction. They reported that the absorbable sponges are worthy for aiding in oral surgery for two reasons: first, as a hemostatic agent, and second, to eliminate “dead space” [22].

The evaluated gelatin-based biodegradable hemostatic sponge with a micro/nanoporous structure was prepared in our previous study [19], which in vitro evaluation showed its hemostatic effect, biocompatibility on human fetal foreskin fibroblasts (HFFF2) cells and red blood cells (RBCs). The consequences of the previous study showed a porous arrangement with micrometric and nanometric pores, flexibility, a two-week range for degradation, and an ability to absorb blood 35 times its weight in vitro. It presented lower blood clotting times (BCTs) (243.33 ± 2.35 s) and a lower blood clotting index (BCI) (10.67 ± 0.004%) compared to two commercial sponges (GELITA-SPON and Spongostan^®^), which showed its ability for faster coagulation and good hemostatic function. In addition, a pilot clinical study with 10 patients who required the extraction of two teeth were tested for absorption of blood and the prepared sponge showed a better ability of blood absorption for the produced sponge (*p*-value = 0.0015) compared with the control [19].

In this study, the produced sponge showed better blood absorption clinically than the control commercial sponge. In our previous study, the sponge showed the ability to absorb blood 35 times its weight. It should be kept in mind that the sponge prepared in this study has a high ability to trap blood cells because of its high porosity [19]. On the other hand, swelling percentage is an important characteristic for hydrophilic and porous materials. According to our results in the previous study, the newly developed sponge shows a high swelling percentage of 3500%. According to reports, high swell ability can improve the homeostasis of sponges by concentrating clotting factors in the sample [33]. Some other researchers also reported the production of porous sponges with the ability to absorb blood 10 times their weight [34].

It can be noted that the prepared sponge traps blood cells due to its high porosity (the microporous structure that matches better with the micrometric size) [19]. Swelling percentage is a specific possession for hydrophilic and porous materials. The newly developed sponge presented a high swelling percentage of 3500%. Based on the reports, high swelling capabilities can improve the hemostasis of sponges by concentrating clotting factors within the sample [33]. Some other researchers testified porous sponges with the ability to absorb blood 10 times its weight as well [34] and 60 times its weight [35]. However, these were not dental sponges. There were no in vitro reports about newly developed dental sponges to compare with our results.

The large surface area of the gelatin sponge can cause the accumulation of platelets during wound closure, and as an absorbable sterile dressing, it provides a mechanical matrix for clot formation and facilitates the formation of a stable clot; therefore, it can reduce the amount of bleeding after tooth extraction [28]. The study of Kim et al. showed that placing an absorbable gelatin sponge in the extraction cavity was a very useful method to prevent postoperative bleeding problems [36]. Tropp et al. showed that sponge Gelfoam is an effective local hemostatic agent after tooth extraction [37].

Based on our outcomes in this study, post-operative bleeding was significantly reduced with test group than control side. The average pain decreased significantly over time in both groups and the reduction of the pain was significantly higher for the test group (*p* < 0.05). Our results for bleeding and pain are substantiated by the literature, thus reaffirming the excellent hemostatic properties of gelatin sponges. Piri et al. showed that the use of a gel tampon in the dental cavity after tooth extraction surgery can reduce the amount of bleeding and pain intensity of patients [28]. Ehab et al. in a randomized controlled trial study for the first time investigated the effects of Alvogyl sponge versus absorbable gelatin sponge as palatal wound dressing and observed that the results of Alvogyl were comparable to absorbable gelatin sponge in terms of postoperative bleeding, also, reduces pain, hemostasis and re-epithelialization properties [38].

A main outcome throughout our study was a case of dry socket in one of our control groups. However, the dry socket did not observe on the extraction socket augmented with the test side of the same subject. It may show the preventive effect of prepared sponge on the occurrence of dry socket, which for approval needs more studies. In a clinical trial, Ghavimi et al. which was conducted on the third molars of the mandible, the results showed that the incidence of dry socket in the control group was (77.8%) and in the experimental group (22.2%), respectively [19]. Wang et al. in a study compared the use of gel tamp (colloidal silver gelatin sponge) with absorbable gelatin sponge with a group without substance (sterile gauze) in the dental cavity after tooth extraction that both showed a significant effect on reducing dry socket [39]. Cai et al. conducted a study on 672 extracted teeth, that showed a significant effect of reducing post-surgical complications (bleeding, pain, infection, swelling, and dry mouth) [21]. It should be noted that the mentioned studies have investigated the effectiveness of sponges compared to the control groups without any substance.

### Trial Strength and Limitations

The main strength of the current clinical trial is the split-mouth project. The split-mouth plan has the important advantage of eliminating intersubject variability from the considered treatment effects. Owing to the dental sponge being locally placed and sutured into the socket, the carry-over effect of the split-mouth strategy is minimal.

One major disadvantage of the study is that it was examined for a short period. A longer time edge assessment of the same might show more accurate consequences.

Even though the split-mouth design is a common and standard design in oral health research, however, pitfalls of the design have been reported and many clinicians are not aware of the potential problems with the split-mouth design.

Although gelatin-based hemostat materials aid the hemostasis without the use of sutures, ligatures, or cautery, their usage is not deprived of risk. Gelatin is made from bovine or porcine sources and may contain some trace biological derivatives that may encourage immunologic responses in some persons.

## 5. Future Perspectives

In order to investigate and better understand the effectiveness of the newly developed hemostatic sponge, future studies can be directed with a larger sample size and in patients with co-morbidities, such as diabetes. The addition of gelatin to other biopolymers may also show promising helpful properties in controlling blood loss and other tooth extraction complications. So, the development of a novel gelatin-based hemostatic agent is necessary in the dentistry field.

## 6. Conclusions

The newly developed sponge material presented great capabilities in bleeding control, pain reduction and avoiding the occurrence of dry socket. These properties can aid to improve sponge functions in the surgical application in dentistry or the other fields of medicine. In this study, it was observed that the total weight of absorbed blood was higher in the control group, which indicates a better ability to absorb blood for the newly developed sponge (teat group). In general, it can be said that gelatin is a safe substance and is easily available. Furthermore, each new gelatin-based homeostatic agent has its own advantage(s) and the need to develop new gelatin-based materials is still relevant.

## Figures and Tables

**Figure 1 jcm-12-04578-f001:**
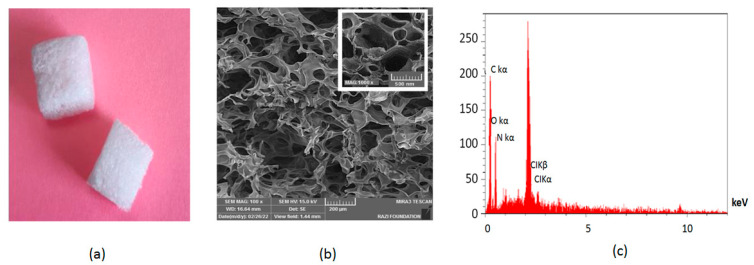
The produced sponge (**a**). The SEM image of the sponge (**b**) and the EDX analysis (**c**). Adopted from [19], which is licensed under an open access Creative Commons CC BY 4.0 license.

**Figure 2 jcm-12-04578-f002:**
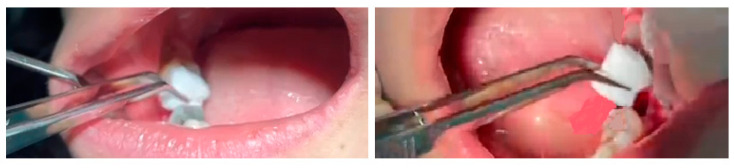
The sponge insertion process.

**Figure 3 jcm-12-04578-f003:**
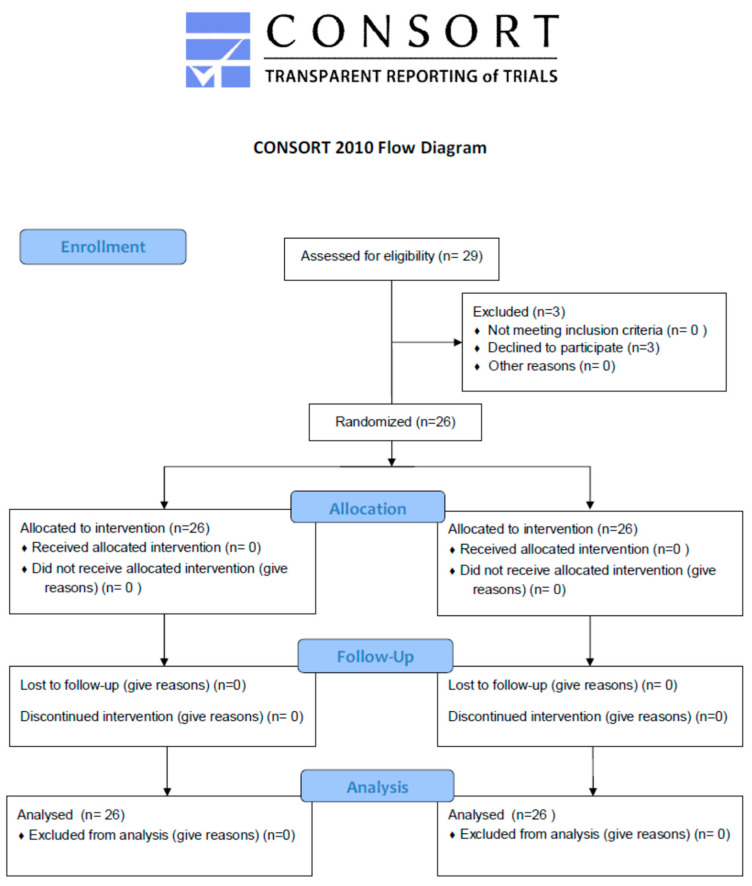
CONSORT flow diagram of the study.

**Table 1 jcm-12-04578-t001:** The demographic details of study subjects.

Variable	Category	Mean ± SD/n(%)
Age	-	31.3 ± 4.02
Gender	Male	12
	Female	14

**Table 2 jcm-12-04578-t002:** Comparing the frequency of gauze numbers in two studied groups.

Number of Gauzes		Control Group	Test Group	*p*-Value
2	Number	6	7	0.845
	Percent	22.2%	25.9%	
3	Number	18	16	
	Percent	66.7%	59.3%	
4	Number	3	4	
	Percent	11.1%	14.8%	
Total	Number	27	27	
	Percent	100.0%	100.0%	

**Table 3 jcm-12-04578-t003:** The results of bleeding for both groups.

Time (h) after Tooth Extraction	Control Group (Mean)	Test Group (Mean)
within 1	1.11 ± 0.23	1.90 ± 0.45
within 1–4	0.23 ± 0.42	0.37 ± 0.31
After 4	0.03 ± 0.21	0.21 ± 0.40

**Table 4 jcm-12-04578-t004:** The average pain over time in test and control groups.

Time (h)	Control Group (Mean)	Test Group (Mean)
12	4.0741 ± 0.71	3.0056 ± 0.80
24	4.4259 ± 0.45	2.0815 ± 0.66
48	2.4630 ± 0.78	1.0667 ± 0.78
72	1.9630 ± 0.65	1.0041 ± 0.57

## Data Availability

The raw/processed data can be shared by request from the corresponding author.

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
