# Peer review of "Efficacy of a New Hemostatic Dental Sponge in Controlling Bleeding, Pain, and Dry Socket Following Mandibular Posterior Teeth Extraction—A Split-Mouth Randomized Double-Blind Clinical Trial"

_jcm, 2023, doi:10.3390/jcm12144578_

Round 1
Reviewer 1 Report
The manuscript is well written and the topic is interesting but there are some points that can be improved:
add, if possible, a paragraph to described in detail the surgical procedure and eventually add photos of patients and procedures;
explain more in detail the results of the statistical analysis creating a paragraph appropriated.
minor revision of the english language
Author Response
The manuscript is well written and the topic is interesting but there are some points that can be improved:
Add, if possible, a paragraph to described in detail the surgical procedure and eventually add photos of patients and procedures;
Explain more in detail the results of the statistical analysis creating a paragraph appropriated.
Comments on the Quality of English Language
Minor revision of the English language
Reply: Thanks for your email. We added some new details based on your comments. The English writing was also checked and corrected comprehensively.
Reviewer 2 Report
Overall study is fine but I have few concerns:
1. What is the rationale of the study, why this material has been synthesized, and what will be the toxicity profile of the synthesized material?
2. Is the efficiency of the synthesized material more than the marketed one?
3. Sentences are repetitive e.g. first para of intro line 1 and line 8.
4. What is meant by the expected date of trial and it's only for 1 day?
5. There are two figures with the same number Figure 1
6. In the consort diagram, the right and left boxes depict the same information
English needs to be improved, rationale needs to be added with some application as compared to what is already marketed.
Author Response
Overall study is fine but I have few concerns:
Thanks for your valuable comments.
- What is the rationale of the study, why this material has been synthesized, and what will be the toxicity profile of the synthesized material?
Reply: This new material was synthesized as a new hemostatic dental sponge in controlling bleeding, pain and dry socket following teeth extraction. It has the efficiency more than two conventional commercial dental sponges. The toxicity tests have been done in our previous study (A Biodegradable Flexible Micro/Nano-Structured Porous Hemostatic Dental Sponge. Published in Nanomaterials, 2022; 12(19):3436) and the results summary has been mentioned in the current manuscript in the discussion section. Please check. We highlighted them.
- Is the efficiency of the synthesized material more than the marketed one?
Replay: In our previous study (A Biodegradable Flexible Micro/Nano-Structured Porous Hemostatic Dental Sponge. Published in Nanomaterials, 2022; 12(19):3436), a comparison of the two conventional commercial brands was done in a laboratory and a clinical pilot (with a small number) and the performance of the manufactured sponge was better compared to both brands. A summary of the results of the previous study was added to the article. Also, in the current clinical study, the manufactured sponge performed better compared to the commercial brand.
- Sentences are repetitive e.g. first para of intro line 1 and line 8.
Replay: It was corrected. Thanks.
- What is meant by the expected date of trial and it's only for 1 day?
The study is a clinical trial. For registration of a clinical trials these two dates (Expected recruitment start date and Expected recruitment end date) is needed to mention. It means the start date of the study steps (preparing materials, preparing patients, the main clinical process) and the end date of the study (ending of the main clinical process, concluding, statistical analyzing, writing the final report and so on). Besides, the trials is designed for 5 days (considering the patients controlling about any signs and symptoms of infection or dry socket) and not 1 day.
- There are two figures with the same number Figure 1
Replay: It was corrected.
- In the consort diagram, the right and left boxes depict the same information
Replay: The study is a split-mouth randomized clinical trial. Then, one side (left) is related to the test group and the other side (right) is related to the control group.
Comments on the Quality of English Language
English needs to be improved, rationale needs to be added with some application as compared to what is already marketed.
Reply: The English writing was checked and corrected comprehensively. Also, some new data were added the better efficiency of the prepared sponge comparing the conventional commercial ones.
Round 2
Reviewer 2 Report
The correction need to be incorporate in manuscript also
Fine
Author Response
The correction need to be incorporate in manuscript also.
Thanks for your valuable comments. We inserted all correction in the manuscript as well. Thanks again. The new modifications have been highlighted in green color.
The previous corrections are in yellow highlights and the English related corrections are in blue color.